# Novel Spatial Approaches to Dissect the Lung Cancer Immune Microenvironment

**DOI:** 10.3390/cancers16244145

**Published:** 2024-12-12

**Authors:** Idania Lubo, Sharia Hernandez, Ignacio I. Wistuba, Luisa Maren Solis Soto

**Affiliations:** Department of Translational Molecular Pathology, The University of Texas MD Anderson Cancer Center, Houston, TX 77030, USA; ilubo@mdanderson.org (I.L.); sdhernandez@mdanderson.org (S.H.); iiwistuba@mdanderson.org (I.I.W.)

**Keywords:** immune profiling, spatial cellular analysis, lung carcinoma

## Abstract

Lung cancer is a deadly disease. Over recent decades, a better understanding of the biological mechanisms implicated in its pathogenesis has led to the development of targeted therapies and immunotherapy, resulting in improvements in patient outcomes. To enhance our understanding of lung cancer tumor biology and advance towards precision oncology, we need a comprehensive tumor profile. In recent years, novel in situ spatial multiomics approaches have emerged, offering a more detailed view of the spatial location of tumor and tumor microenvironment cells. These advancements in in situ profiling may unveil further molecular and immune mechanisms in tumor biology that will lead to the discovery of biomarkers for treatment prediction and prognosis. In this review, we provide an overview of current and emerging pathology-based approaches for spatial immune profiling in lung cancer.

## 1. Introduction

Lung cancer is a deadly disease, with the highest rates of mortality worldwide [1]. It is classified by its morphological and immunohistochemical features as small-cell lung carcinoma (SCLC) (~14% of cases) and non-small cell lung carcinoma (NSCLC) (more than 80%), which includes lung adenocarcinoma (~62%) and squamous cell carcinoma (~24%) [1,2,3,4]. Lung cancer subtypes have distinct molecular alterations, disease evolution, and heterogeneity at molecular, cellular, and immune levels [5,6,7]. Over recent decades, a better understanding of the biological mechanisms implicated in lung cancer pathogenesis has led to the development of targeted and immunotherapy approaches that have resulted in improvements in patient outcomes [8,9,10,11,12,13,14,15,16,17,18,19].

Importantly, in the last decade, the scientific breakthrough of immunotherapy against immune checkpoint inhibitor (ICI)–programmed cell death protein 1 (PD-1)/programmed cell death ligand 1 (PD-L1) and recent progress in the understanding of cancer immune response have led to the approval of immunotherapy for early and advanced NSCLC stages [14,20,21,22,23]. However, tumor cells can develop immune resistance mechanisms including genetic or epigenetic changes (tumor-intrinsic features) and changes in the surrounding tumor microenvironment (TME) (tumor-extrinsic features). Furthermore, patients’ clinico-biological features beyond the tumor and its TME (host features) play an important role in response to immunotherapy [24,25,26,27]. The growing understanding of the tumor’s immune profile and its dynamic changes during tumor progression and therapy has led to the development of novel immunotherapy approaches for NSCLC, including novel immune-checkpoint targets, adoptive cell therapy (chimeric antigen receptor T cells or natural killer (NK) cell therapy, T cell receptor gene therapy, and tumor-infiltrating lymphocyte therapy), and therapeutic vaccines [27,28]. These therapies are being studied in combination with standard treatment or novel targeted approaches, including antibody–drug conjugates or molecular targeted agents [28,29,30,31].

Several recent methodological advances have been made to understand the tumor biology of SCLC, including integrative multiomic approaches in tumor samples, cell lines, and genetically engineered mouse models. These studies allow to categorize SCLC tumors on the basis of their transcriptomic profiles, unveiling potentially targetable tumor vulnerabilities that can be exploited to overcome resistance. Currently, novel targets and strategies for precision therapy in SCLC are being investigated in clinical trials, including targeted kinase inhibitors, monoclonal antibodies, angiogenesis inhibitors, DNA-damaging agents, and antibody–drug conjugates [32,33,34,35,36].

To enhance the understanding of lung cancer tumor biology towards precision oncology, a comprehensive tumor profile is necessary. The evaluation of tumor cells, TME cells, and surrounding tissue at different stages of tumor development can lead to the refinement of phenotypic, molecular, and immune profiles to understand tumor heterogeneity and tumor plasticity, which are essential to designing better strategies to improve outcomes in lung cancer patients. This deep profiling is now possible thanks to recent advancements in genomics, imaging, and computational biology, including fundamental techniques such as immunohistochemistry and in situ hybridization, to high-throughput gene expression methods such as RNA sequencing, with the ultimate development of modern spatial multiomic techniques, which combine gene or protein expression analysis with spatial information, providing novel molecular insights of the complexities of tumor architecture at cellular and subcellular levels and its implications in cancer evolution [37].

In this review, we provide an overview of tumor-associated immune biomarkers and insights into current and emerging pathology-based approaches for spatial immune profiling of lung cancer, including spatially resolved proteomic assays and next generation sequencing (NGS)-based and images-based technologies. Lastly, we briefly discuss other spatial profiling methods such as metabolomics and 3D technologies.

## 2. Current and Emerging Tumor Biomarkers for Immunotherapy

Certain intrinsic features of lung cancer tumors have served as biomarkers that can predict response to immunotherapy, including tumor mutational burden and PD-L1 expression in malignant cells, both of which are currently approved by the United States Food and Drug Administration (FDA) as standard biomarkers for ICI treatment in patients with NSCLC [12,38]. High tumor mutational burden (>10 mut/Mb) is associated with favorable response to ICIs. Tumor mutation load is correlated with neoantigen load, which accounts for the immunogenicity of tumors; therefore, higher levels drive the recruitment and activation of immune cells (T cells) that histologically correspond to an inflamed or hot phenotype [39,40]. Although PD-L1 is expressed in tumor and immune cells, the biomarker is usually assessed by single chromogenic immunohistochemistry in tumor tissues, and a higher tumor proportion score (percentage of tumor cells with complete circumferential or partial membrane expression of PD-L1) is associated with a better response to ICIs [41].

The presence of distinct tumor mutation profiles is another important lung cancer feature, in addition to PD-L1 expression and TMB. Certain oncogenic drivers, such as mutation of epidermal growth factor receptor (*EGFR*), are predictive of low response to anti-programmed cell death protein 1 therapies. Furthermore, these tumors usually harbor low tumor mutational burden and diminished tumor T cell infiltration. Lung adenocarcinoma with co-mutations of Kirsten rat sarcoma viral oncogene homologue (*KRAS*) and loss of function of *STK11/LKB1* also reduce tumor T cell infiltration and are predictive of relapse and poor outcome in patients who have undergone anti-PD-1 treatment [42,43].

The TME of lung cancer has been under exhaustive investigation to identify biomarkers of response to ICIs. Translational studies have increasingly been integrating tumor profiling and spatial analysis into immunotherapy clinical trials for various tumor types including lung cancer [44]. Overall, higher numbers of CD4+ and CD8+ cells have been shown to be predictive of better outcomes in patients with advanced NSCLC treated with ICIs [45,46] as well as the presence of tertiary lymphoid structures [47,48]. In patients with resected NSCLC who have been treated with neoadjuvant immunotherapy, T cells, cytotoxic T cells, memory cells, NK cells, NK cell-like T cells, and T cell repertoire have been associated with major pathological response [49,50,51]. In the phase 2 randomized NEOSTAR trial (NCT03158129) conducted to understand the effect of ICIs (neoadjuvant nivolumab or nivolumab plus ipilimumab) in resectable early-stage NSCLC, mIF and other techniques demonstrated that compared to pre-therapy tumor samples, post-tumor samples demonstrate increases in tumor T cell infiltration and immunologic memory [49]. Likewise, correlative analysis of the Lung-MAP S1400I Phase 3 Randomized Clinical Trial (NCT02785952), conducted to study the benefit to adding ipilimumab to nivolumab in patients with advanced squamous cell lung carcinoma, chemotherapy-pretreated [52], showed that active immune infiltration, measured by multiplex immunofluorescence (mIF) and gene expression profiling in baseline biopsies, was linked to a benefit from ICIs. Furthermore, exceptional responders were associated to higher infiltration of cytotoxic T cells and memory cytotoxic T cells; also, the spatial patterns of these immune cells were characterized for having higher spatial proximity to malignant cells in these patients [53].

Some studies have complemented the investigation of immune cell densities and percentages by evaluating the association of spatial patterns of immune cells and immunotherapy outcomes. In 42 NSCLC patients who were treated with ICIs following recurrence after surgery, non-responders had higher percentages of Treg cells; in addition, Tregs had higher proximity to monocytes and CD8+ cells in non-responders, while macrophages had higher proximity to HLADR+ tumor cells in responders [54]. Information about the color, shape, and texture of immune cells and their spatial location patterns has also been extracted using novel computational approaches with H&E-stained images of pre-treatment tumor samples from patients with NSCLC treated with chemotherapy or immunotherapy [55].

Moreover, integration of multiomic technologies using DNA and RNA approaches techniques, such as whole exome sequencing, whole genome sequencing, targeted sequencing, RNA sequencing, and single-cell RNA sequencing, have been used to explore the high complexity of the TME, and can further aid in biomarker discovery. For instance, single-cell sequencing in early-stage lung adenocarcinoma has identified KRT8+ alveolar intermediate cells as potential precursors of malignancy, with the integration of whole exome and spatial transcriptomics data [56]. Additionally, single-cell RNA sequencing of B cells, and their receptor, revealed an increase in B cells within the TME of lung adenocarcinoma with a notable maturation towards plasma cells, especially among smokers. The abundance of plasma cells was associated with improved survival and better response to immunotherapy, suggesting the use of the signature of plasma cells as a potential biomarker for response to anti-PD-1/PD-L1 therapy [57]. Furthermore, a reduction in cytolytic activity related to cell mediators of antitumor immunity was identified using single-cell sequencing and validated by flow cytometry and mIF [58].

The spatial location of TME cells is fundamental to the investigation of interactions among cells and to the inference of functional states that can have an impact on tumor fate; geospatial analysis is a field with growing relevance, and can be performed considering different TME cell types, such as immune cells, fibroblasts, vessels, and nerves [59] (Figure 1). Advances in single-cell RNA sequencing technologies combined with state-of-the art computational approaches have provided high-dimensional transcriptomic information on single cells, allowing us to classify and investigate similarities between them, infer cell–cell communication, and study cell plasticity, heterogeneity, and functional states [60]. In addition to characterizing cell densities and percentages of immune cells, spatial analyses can be performed to study proximity among cells of interest using nearest neighbor distances or categorize spatial patterns using neighborhoods or communities. Current advances in in situ profiling that account for spatial location and histological context using high-throughput technologies and computational approaches for data analyses may unveil further molecular and immune mechanisms in tumor biology that will lead to the discovery of biomarkers for treatment prediction and prognosis [61].

## 3. Novel Spatial Multiomic Approaches to Studying the Tumor and TME

In recent years, the development of high-throughput bulk proteomics such as mass spectrometry (MS) or reverse phase protein array (RPPA) have allowed to study and quantify thousands of proteins and unveiled several aspects related to post-translational modifications, protein–protein interaction, signaling pathways, and the discovery of biomarkers that can be used in translational or clinical studies; these discoveries, when integrated with genomic, transcriptomic, epigenomic, and metabolomic analysis, are enhancing our understanding of lung tumor biology and the evolution of this disease [62,63,64]. Currently novel in situ spatial multiomic approaches have emerged that offer a more detailed view of tumors’ topographical organization and interaction with the TME. Below, we describe the more widely used techniques and how they enhance our understanding of tumor biology and immune response.

### 3.1. High-Plex Protein Profiling

Spatially resolved proteomic assays provide critical information about tissue cell-to-cell interactions and cell phenotypes through the characterization of multiple protein biomarkers using different methods [65] (Figure 2).

Multiplex immunofluorescence (mIF) relies on multiple fluorescence signal expression using tyramide signal amplification and multi-spectral imaging to visualize single cells with one or multiple co-expressing targets; however, it has a limited assay capability of six to nine targets [66]. A review of this assay’s capabilities was previously presented [67].

Cyclic immunofluorescence staining and imaging are high-plex technologies that offer single-cell resolution of 10–100 markers, overcoming the limited capability of mIF assays [65]. These cyclic imaging technologies can be based on target antibodies conjugated with oligonucleotides and fluorophores (PhenoCycler-Fusion, Akoya Bioscience) [68], sequential cycles of immunofluorescence antibodies (e.g., COMET, Lunaphore) [66], and cyclic chromogenic staining (Multiplexed Immunohistochemical Consecutive Staining on Single Slide) [68].

Single-step imaging technologies use either heavy metal-tagged probes (imaging mass cytometry or multiplexed ion beam imaging) or fluorophores (Orion, Rarecyte), resulting in the visualization of single-cell multiple markers with one-step staining [61]. As an example of the utilization of this approach in lung cancer, in one study, highly multiplexed imaging mass cytometry was used to analyze 416 primary lung adenocarcinoma samples from treatment-naïve patients, most of them early stage (I–II, 87.74%). In addition to tumors and endothelial cells, they profiled 14 immune cell types. The study evaluated the correlation between cell frequencies, their interactions with clinical data, and the link between survival and cellular phenotypes in the TME, finding that, independently from tumor architecture, the spatial location of the cells predicts survival [69].

Multicellular protein profiling: Digital spatial profiling (DSP) technology (GeoMx DSP, NanoString/Bruker) and the Visium protein assay (10x Genomics) offer high-plex protein profiling of regions of interest (ROIs) and spatial resolution [70]. The protein assay of the Visium platform profiles 35 proteins, along with the gene expression panel of the standard assay [71]. GeoMx DSP performs protein profiling through the collection and quantification of oligonucleotides attached to up to 570 primary antibodies (IO Proteome Atlas (IPA)) [72], and the ROI selection is aided by the visualization of biological compartments with standard fluorescence [73]. Some studies have been conducted in lung cancer using spatial multiomic analysis such as the high-plex digital spatial profiling (DSP). For example, in samples from patients enrolled in a phase II multicenter clinical trial (NCT03838848) [patients with advanced NSCLC treated with anti-PD-L1/CTLA-4 bispecific antibody (KN046)], researchers analyzed the impact of intratumoral heterogeneity using spatial proteomic and transcriptomic DSP and found spatially defined tumor and stromal signatures scores that could predict clinical response [74]. Similarly, using spatial protein profiling, a study of 67 advanced NSCLC samples from patients treated with anti-PD-1 therapy showed expression of VISTA and CD127 in the tumor compartment as markers of immunotherapy resistance, while high CD56 and CD4 were associated with both overall and progression-free survival [75].

### 3.2. Spatial Transcriptomic Analysis

Several assays have been recently developed to evaluate gene expression, considering the histological topography of tumor cells and TME with multicellular, cellular, and subcellular resolution. This resolution can be applied to create a molecular atlas of different cell types with distinct gene programs and evaluate the molecular state of these cells and their organization in distinct histological niches [76,77]. Overall, there are two main spatial transcriptomics analysis methods: next-generation sequencing (NGS)-based and image-based technologies [78,79,80] (Table 1). With both, pre-assessment of tissue RNA integrity is necessary to ensure high-quality data [81].

NGS-based methods allow us to map the sequencing data obtained by NGS in tissue; different approaches have been developed, including GeoMx DSP, oligonucleotide-based spatial barcoding on slides (Visium Spatial Gene Expression assay, 10x Genomics), and spatial barcoding on a bed array (Slide-seq, SeqScope, and Stereo-seq). These techniques are considered spatial multicellular methods since they do not allow cellular and subcellular resolution, thus limiting their application to studies of cell neighborhoods and cell-to-cell interactions; however, they allow us to profile the whole transcriptome and can be used for discovery studies [82].

Image-based methods allow the visualization of single molecules in histological tissue and rely on two major technologies, in situ hybridization and in situ sequencing [82]. In situ hybridization technologies allow the visualization of specific target probes in tissues and include single-molecule fluorescence in situ hybridization (smFISH) [83], sequential fluorescence in situ hybridization (seqFISH) [84], multiplexed error-robust FISH (MERFISH, Vizgen) [85], molecular cartography [86], spatial molecular imager (SMI) (CosMx, NanoString/Bruker) [87], and In Situ Gene Expression (Xenium, 10x Genomics) [88]. In situ sequencing technologies are based on mRNA sequencing in intact tissues and include STARmap, Instaseq, FISSEQ, and HybISS [78,89]. Both technologies enable cellular and subcellular resolution but are limited to target panels [82] (Figure 3).

#### 3.2.1. Spatial Multicellular Methods

Spatial multicellular methods provide spatial data without achieving single-cell resolution. Here, we describe three commercially available platforms that are available for evaluating formalin-fixed, paraffin-embedded (FFPE) tissue.

The spot-level-resolution Visium Spatial Gene Expression assay is a probe-based, whole-transcriptomic spatial transcriptomics assay that measures transcriptome profiles and co-detects proteins in spots within a 55 μm diameter, spaced 100 μm apart from center to center, leaving gaps between spots [90,91]. Recently, a new high-definition format has been developed with a reduced spot size of 2 μm squares, without gaps. In both versions, mRNA targets are captured by probe ligation and bind to the Visium slide after tissue permeabilization. The mRNA captured at each spot or square, along with unique barcode sequences, corresponds to specific XY coordinates that are used to map transcriptomic information to histological slides. Most investigators use H&E-stained slides; however, the assay is also compatible with immunofluorescence [56,92]. One key advantage of the whole-transcriptomic spatial transcriptomics assay is that it captures all of the mRNAs transcribed in each region, enabling the analysis of whole tissues and assessment of up to 18,000 genes. The alignment of gene expression data and histological features can enhance the investigation of intratumoral heterogeneity and transcriptional programs associated with histological morphologies pertinent to lung cancer pathogenesis, including distinct analysis in tumor tissue and adjacent normal structures, aimed at biomarker discovery [90,91]. Distinct features within tissues allow the comparison of gene expression and histological characteristics [93]. The main disadvantage of the assay, even with the resolution improvement in the high-definition version, is that it still lacks single-cell resolution, making it a multicellular assay. It has also been proposed that during tissue permeabilization, probes spread into nearby spots, producing an effect known as “spot swapping” [94].

*Digital spatial profiling technology (GeoMx DSP)* is a high-plex platform that enables spatial gene Whole Transcriptome Atlas profiling in FFPE tissue. Slides are processed with in situ hybridization probes attached to photocleavable oligonucleotide tags plus mIF biomarkers to select ROIs [95,96]. The tags are released, quantified, and remapped to the tissue. The ROIs can be selected in different shapes and sizes (up to 660 × 785 µm) [97,98]. These characteristics allow the platform to act as a discovery tool, profiling around 18,000 genes in multiple selected ROIs from whole tissue sections, maintaining the spatial organization of the cells in the tissue and offering insights into tissue heterogeneity [99].

The *spatial enhanced resolution transcriptomics sequencing assay (Stereo-seq, STOmics)* is a spot-level technique that offers resolution at the nanoscale level (220 nm spot size) with a wide field of view (up to 13 × 13 cm) and the ability to collect several data locations per cell, capturing the expression of every gene present in the sample (whole-transcriptome profiling) [100,101]. The main advantage of Stereo-seq is that it allows the platform to be species-agnostic; therefore, it has the capability to also extract information on the microbiome [101,102,103]. This technique has been used in one study of NSCLC, with integration of single-cell RNA sequencing, revealing diverse phenotypes and gene expression profiles of cancer-associated fibroblasts (CAFs). Notably, a subcluster of myofibroblastic CAFs, POSTN+, was enriched in advanced tumors, exhibiting signatures linked to extracellular matrix remodeling, tumor invasion, and immune suppression, and hence associated with poor prognosis [104].

#### 3.2.2. Spatial Single-Cell Sequencing Methods

Spatial single-cell sequencing approaches allow the visualization of single molecules within the tissue. Unlike traditional technologies, which enable in-depth studies of cellular heterogeneity but lose the spatial location, single-cell spatial approaches can preserve this valuable feature.

*RNAScope (ACD Bio)* technology employs an advanced in situ hybridization technique with a double-Z probe design, minimizing noise signaling and allowing for the detection, visualization, and localization of RNA or other targets of interest (using custom probes) with subcellular and cellular resolution. This platform can be used with single and dual chromogens and mIF. Using mIF, the platform is currently able to evaluate up to 12 probes [105].

The In Situ *Gene Expression technique (Xenium)* uses a microscopy-based readout with padlock probe technology and gene-specific barcodes that bind to the target transcript [88]. The process is followed by probe ligation and enzymatic amplification. Currently, their panels can target up to 5000 genes, but the technology also allows panel customization, including sequences to track engineered cells [88,106]. The assay offers gene-specific ligation with strong and bright signals for imaging in a larger imageable area (12 × 24 mm) [100,101,107].

*Spatial Molecular Imager (SMI) technology (CosMx)*: This assay involves tissue permeabilization, probe hybridization, and slide assembly insertion on an automated molecular imaging platform, as well as a 64-bit multiplexing technique that enables the simultaneous detection and quantification of RNA and protein for more than 1000 targets at the single-cell resolution level. With SMI, four slides can be scanned at once in a scan area of 16 × 375 mm^2^ on each tissue slide [87].

*Multiplexed error-robust FISH (MERFISH)* enables single-cell resolution by applying combinatorial FISH labeling with multiplexed, error-robust encoding schemes, with successive imaging rounds to detect transcripts [85]. Currently, this platform offers custom panels of up to 1000 genes [108].

In lung cancer research, several studies have used these spatial technologies to precisely locate cells identified through single-cell RNA sequencing and associate them with histological features that are relevant to the pathogenesis of lung adenocarcinoma [56,57]. Although biomarker gene signatures of lung cancer have not yet been used in clinical settings, compartment-specific markers and molecular phenotypes obtained from a transcriptional analysis of groups of cell populations inside the space of intricate tissues may allow us to not only gain a better understanding of the factors that affect patient outcome [109] but also predict response to immunotherapy [110].

## 4. Other Spatial Profiling Methods

### 4.1. Metabolomics

Metabolic reprogramming significantly influences the progression of different types of cancer; these specific metabolic alterations are still not fully understood, limiting their clinical application [111]. Therefore, to overcome this lack of knowledge, several studies are undergoing in this area. For example, in lung cancer, high levels of glycine, glycolate glyceric acid, creatinine, aminomalonic acid, and oxalate were identified as risk factors in a study that used gas chromatography–mass spectrometry to identify plasma metabolites. Oxalate was found to be a unique metabolic risk factor and lactate dehydrogenase A, an enzyme related to oxalate metabolism, was associated with poor prognosis [112]. In this context, RPPA technology, integrated with different analytic platforms, such as mass spectrometry, metabolomics, transcriptional profiling, DNA sequencing, and epigenomics, has proven to be particularly useful [113]. In patients with lung adenocarcinoma, mitochondrial proteins related to fatty acid oxidation, antioxidant response, and oxidative phosphorylation, identified with RPPA, showed being an independent predictor of survival, encouraging the use of mitochondrial activity as target of therapy [114]. The study of the metabolic programs in the tumor and TME is an emerging area in immune-oncology research because of these programs’ impact on cells’ function [115,116]. Multiplex or high-plex protein profiling can be used to study these programs within the context of tissue heterogeneity using antibodies that target key metabolic pathways, spatial transcriptomics to study gene signatures or pathways related to metabolic processes, and technologies that allow us to visualize metabolites using mass spectrometry imaging, including matrix-assisted laser desorption/ionization, desorption electrospray ionization, and secondary ion mass spectrometry, which have allowed the acquisition of spatial metabolite features [117,118].

### 4.2. 3D Technologies for Histological and Molecular Profiling

One of the challenges in immune profiling using 2D approaches is the truncated information obtained from the TME, since many cells and structures benefit from 3D investigation to capture relevant microscopic architecture patterns, such as microvascular architecture and cell prolongations from nerves or immune cells; 3D approaches are designed to better understand cell-to-cell interactions and molecular and cellular niches and their dynamics in the tumor and TME [119].

To address this need, there are two main 3D technologies: (a) serial sectioning and 3D in silico reconstruction, in which the sections are used for in situ spatial profiling, where images and molecular information are aligned by means of advanced computational approaches; and (b) non-destructive optical sectioning microscopy, which uses technological advancements in tissue imaging to obtain volumetric information and tissue properties and may infer data from molecular and H&E-stained images, which ultimately allows the integration of diverse cellular states from the tumor and TME, with distinct microarchitecture patterns and cell signaling [77,120,121,122].

## 5. Conclusions and Future Directions

Traditionally, the study of lung cancer has relied on the analysis of conventional hematoxylin and eosin slides and low-plex immunohistochemistry [4]. Nowadays, tumors and their microenvironment are the subject of constant discovery, and the understanding of the TME has been revolutionized by bulk single-cell techniques, further enhanced by spatial transcriptomics data and spatial interactions within tumor cells and TME components.

In addition to the proteome, the transcriptome can be useful for designing new therapies, identifying genes that are impacted by various genomic and epigenomic changes and the detection of large rearrangements of chromosomes [64]. Integrating diverse levels of analysis including not only spatial proteomics and transcriptomics, but also metabolomics, genomics, and epigenomics, can enhance our understanding of the spatial relationship of cells previously identified with single-cell analysis, the correlation with their functional status, and the understanding of different biological pathways. This integration allows us to identify biological changes found in the natural history of the development and progression of lung cancer, enabling the collection of more precise data and hence potential tailored treatments [123].

The use of multiomics techniques can help to address the biological differences between the transcriptome and proteome. Numerous variables, including post-transcriptional regulation, post-translational modifications, protein breakdown, protein–protein interactions, and environmental stressors, can contribute to this discrepancy. Phosphorylation, acetylation, and glycan modification are examples of changes that can affect proteins and impact biological processes including immunological responses, cell signaling, and cancer. Understanding and adjusting these changes could therefore result in the discovery of biomarkers or signatures and novel therapeutic approaches [64].

Since the quality of the data generated from spatial methodologies heavily relies on the experimental design, when selecting a platform to profile lung cancer it is essential to consider the scope of the study, to identify which questions the technology is best suited to answer, and to consider each sample’s inherent heterogeneity. Another important point to consider is time, especially when the biological question comprises the response to treatment and resistance mechanisms [124]. Therefore, it is crucial to rigorously assess every aspect of collection and preparation of samples (preanalytical factors), the methodology of the spatial analysis (resolution, capture area, profiling panel), data generation, and analysis [125]. Furthermore, to obtain clinically significant findings, the integration of multiomics data requires various strategies, including conceptualization, data integration, prediction models, networks, and pathways.

Beyond the study planning and execution, performance evaluation of the platforms is a matter to be considered. In this context, one comparison study showed moderate to modest concordance of three commercial single-cell spatial transcriptomics techniques with matching bulk RNA sequencing, revealing some issues with cell segmentation and probe sensitivity in lung cancer and mesothelioma samples [126]. Another study that compared imaging spatial transcriptomics assays on tissue microarrays serial sections containing tumor and normal tissue found that the platforms evaluated were consistent with orthogonal RNA-seq datasets with different cell segmentation, discovery rates, and degrees of sub-clustering performance for analysis [127]. These individual characteristics of the platforms can become advantages or liabilities for the purpose of our study, and it is crucial to consider them instead of only being guided by the idea of how much data we can produce.

Currently, there are several initiatives to provide best practices for spatial technologies, such as validation and optimization of the assays, and proper normalization strategies, which are also challenging issues since there are no common standards [124,128,129]. Harmonizing the various processes and putting in place quality control measures for the variation in the different tests is essential. These harmonization processes are necessary to guarantee that data produced from several sites can be compared, allowing accurate results [130,131,132]. Also, it is very important to include a histopathology perspective of these approaches, especially when using image-based methods, where it is possible to correlate molecular data with tissue morphology characterization [133]. Moreover, it is essential to also establish workflows with a multidisciplinary team that ensures multiomics data integration with clinical information, including the nuances in responses to ICIs and targeted therapies. Although some assays offer basic tools for analysis, in general, the complexity of data analysis and interpretation rises with the number of biomarkers and gene information obtained from the samples, requiring advanced bioinformatic skills and the use of specific software for each step of the analysis, being a significant limitation for application in clinical settings [133].

In conclusion, our understanding of lung cancer tumor biology has advanced in recent years, thanks to the development of technologies that allow the profiling of tumor and immune cells to identify their unique composition and functional status. Novel multiomic platforms have been developed and incorporated into clinical trials to evaluate tumor heterogeneity, dynamic gene expression, metabolic reprogramming, signaling pathway activation, cell-to-cell interactions, and immune cell programs. A better understanding of lung cancer tumor biology will lead to biomarker breakthroughs and tailored treatments for patients, while future research should focus on longitudinal spatial profiling tracking changes in the immune landscape over the course of disease progression, treatment, and resistance, in addition to spatial effects of metabolic pathways connected to immune cell distribution and niches’ roles within the lung cancer TME.

## Figures and Tables

**Figure 1 cancers-16-04145-f001:**
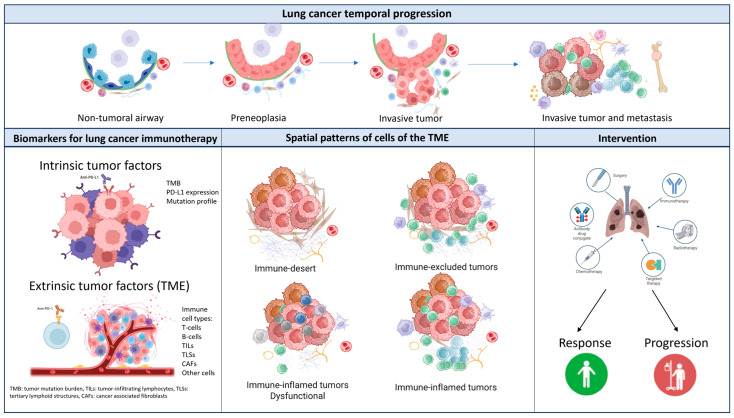
Spatiotemporal features and biomarkers of the tumor and its TME in lung cancer.

**Figure 2 cancers-16-04145-f002:**
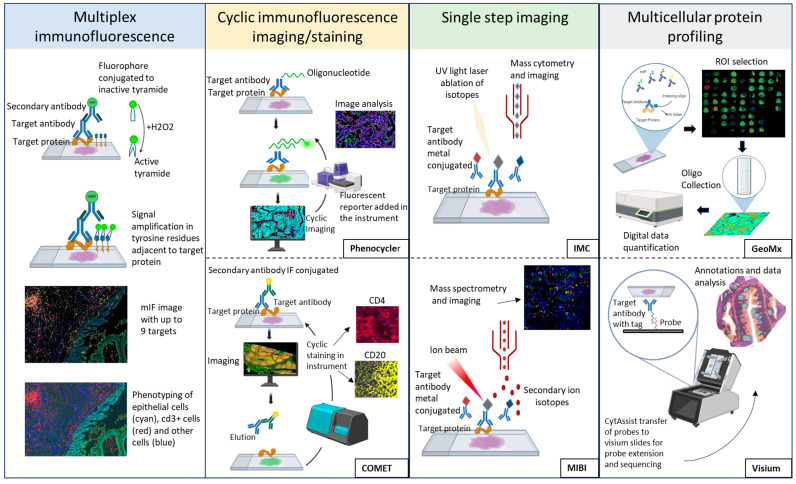
Spatially resolved proteomic assays.

**Figure 3 cancers-16-04145-f003:**
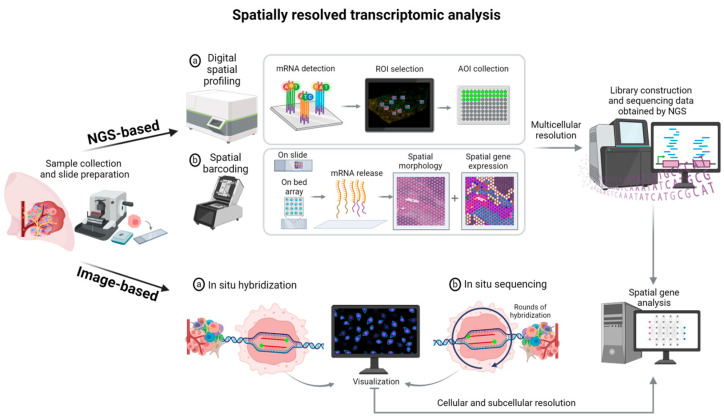
Spatially resolved transcriptomic analysis.

**Table 1 cancers-16-04145-t001:** Characteristics of spatial transcriptomic technologies.

Technology	Resolution	Target	Capture Area	Advantages
Spatial Gene Expression assay (Visium)	Multicellular, spot-level (NGS-based)	RNA and protein, ~18,000 genes	6.5 × 6.5 mm, 11 × 11 mm	Whole transcriptome in whole tissue
Digital spatial profiler (GeoMx DSP)	Multicellular, biological compartments (NGS-based)	RNA and protein, ~18,000 genes	35 × 14 mm ROIs, 600 × 785 µm	Whole transcriptome in distinct biological and histological compartments
Stereo-seq	Cellular and subcellular spot-level nanoscale (NGS-based)	RNA, whole transcriptome	Wide range, up to 13 × 13 cm	Whole transcriptome in whole tissue; species agnostic
RNAscope	Cellular and subcellular (image-based)	RNA, target gene expression up to 12 genes	Whole slide	Visualize and quantify CAR/TCR vectors and cytokines, immune markers, other targets of interest; visualize microbiome
Xenium	Cellular and subcellular (image-based)	RNA, up to 5000 genes	12 × 24 mm	Single-cell resolution; cell-to-cell interaction
CosMx SMI	Cellular and subcellular (image-based)	RNA, up to 6000 genes; protein, up to 64 targets	15 × 20 mm (FOV, 540 × 540 µm)	Single-cell resolution; cell-to-cell interaction
MERFISH	Cellular and subcellular (image-based)	RNA, up to 1000 genes	1 cm^2^	Single-cell resolution; cell-to-cell interaction

## Data Availability

No new data were created or analyzed in this study. Data sharing is not applicable to this article.

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
