# Peer review of "Novel Spatial Approaches to Dissect the Lung Cancer Immune Microenvironment"

_cancers, 2024, doi:10.3390/cancers16244145_

Round 1
Reviewer 1 Report
Comments and Suggestions for Authors
The article "Novel Spatial Approaches to Dissect the Lung Cancer Immune Microenvironment" is a good contribution to the field of spatial proteomics in oncology, and should be of interest to the readership of the journal. The authors have done an excellent job of highlighting the advancements in spatial multiomics technologies and their application in understanding the complex interactions within the tumor microenvironment. The comprehensive overview of current and emerging pathology-based approaches for spatial immune profiling in lung cancer is particularly impressive. This work not only enhances our understanding of tumor biology but also gives novel unsights for the discovery of new biomarkers and therapeutic targets. The integration of spatial technologies to provide a detailed view of the tumor and its microenvironment is a significant step forward in precision oncology. Overall, this article is a valuable resource for researchers and clinicians alike, offering insights that could lead to improved patient outcomes.
Minor points
Discussing the integration of these spatial technologies with classical bulk proteomics approaches would improve the discussion, in line with what was described in this recent paper https://doi.org/10.3390/proteomes11040034
Author Response
Reviewer #1: 
We would like to thank the reviewer for taking the necessary time and effort to review the manuscript. We sincerely appreciate all your valuable comments and suggestions, which helped us in improving the quality of the manuscript.
Comments and Suggestions for Authors:
“ Discussing the integration of these spatial technologies with classical bulk proteomics approaches would improve the discussion, in line with what was described in this recent paper https://doi.org/10.3390/proteomes11040034”
Response: We thank the reviewer for the comments and suggestions. To further discuss the integration of spatial technologies with classical bulk proteomics, we added the following information:
- Line 173-179: “In recent years, the development of high throughput bulk proteomic such as mass spectrometry (MS) or reverse phase protein array (RPPA) have allowed to study and quantify thousands of proteins and unveiled several aspects related to post-translational modifications, protein-protein interaction, signaling pathways and discover of biomarkers that can be used in translational or clinical studies, these discoveries when integrated with genomic, transcriptomic, epigenomic, and metabolomic are enhancing our understanding of lung tumor biology and evolution of this disease [62-64]”.
- Line 388 – 405: “In addition to the proteome, the transcriptome can be useful for designing new therapies, identifying genes that are impacted by various genomic and epigenomic changes and the detection of large rearrangements of chromosomes [64]. Integrating diverse levels of analysis including not only spatial proteomics and transcriptomics, but also metabolomics, genomics and epigenomics, can enhance our understanding of the spatial relationship of cells previously identified with single-cell analysis, the correlation with their functional status and the understanding of different biological pathways. This integration allows us to identify biological changes found in the natural history of the development and progression of lung cancer, enabling the collection of more precise data and hence potential tailored treatments [125].
The use of multi-omics techniques can help to address the biological differences between the transcriptome and proteome. Numerous variables, including post-transcriptional regulation, post-translational modifications, protein breakdown, protein-protein interactions, and environmental stressors, can contribute to this discrepancy. Phosphorylation, acetylation, and glycan modification are examples of changes that can affect proteins and impact biological processes including immunological responses, cell signaling and cancer. Understanding and adjusting these changes could therefore result in the discovery of biomarkers or signatures and novel therapeutic approaches [64]”.
Reviewer 2 Report
Comments and Suggestions for Authors
1. In Section 1, an overview of Novel Spatial Approaches should be provided.
2. In Section 2, clinical research data or real-world case studies should be added to demonstrate the advancements in tumor biomarkers.
3. Sections 3 and 4 need additional examples for support.
4. The content in Section 5 is vague and lacks substance.
Author Response
Reviewer #2: 
We would like to thank the reviewer for taking the necessary time and effort to review the manuscript. We sincerely appreciate all your valuable comments and suggestions, which helped us in improving the quality of the manuscript.
Comments and Suggestions for Authors:
Comment 1: “In Section 1, an overview of Novel Spatial Approaches should be provided”.
Response: We thank the reviewer for this valuable suggestion. An overview of novel spatial approaches was added in the following sections of the manuscript:
- Line 75 – 87: “This deep profiling is now possible thanks to recent advancements in genomics, imaging, and computational biology, including fundamental techniques such as immuno-histochemistry and in situ hybridization, to high-throughput gene expression methods such as RNA-sequencing, with the ultimate development of modern spatial multi-omic techniques, which combine gene or protein expression analysis with spatial information, providing novel molecular insights of the complexities of tumor architecture at cellular and subcellular levels and its implications in cancer evolution [37].
In this review, we provide an overview of tumor-associated immune biomarkers and insights into current and emerging pathology-based approaches for spatial immune pro-filing of lung cancer, including spatially resolved proteomic assays and next generation sequencing (NGS)-based and image-based technologies. Lastly, we briefly discuss other spatial profiling methods such as metabolomics and 3D technologies”.
Comment 2: “In Section 2, clinical research data or real-world case studies should be added to demonstrate the advancements in tumor biomarkers”.
Response: We thank the reviewer for this helpful comment. To address the lack of clinical studies in our manuscript, we added examples in the following sections:
- Line 118 – 130: “In the phase 2 randomized NEOSTAR trial (NCT03158129) conducted to understand the effect of ICIs (neoadjuvant nivolumab or nivolumab plus ipilimumab) in resectable early-stage NSCLC, mIF and other techniques demonstrated that compared to pre-therapy tumor samples, post-tumor samples have increases in tumor T-cell infiltration and immunologic memory [49]. Likewise, correlative analysis of the Lung-MAP S1400I Phase 3 Randomized Clinical Trial (NCT02785952), conducted to study the benefit to adding Ipilimumab to Nivolumab in patients with advanced squamous cell lung carcinoma chemotherapy-pretreated [52], showed that active immune infiltration, measured by multiplex immunofluorescence (mIF) and gene expression profiling in baseline biopsies were linked to a benefit from ICIs. Furthermore, exceptional responders were associated to higher infiltration of cytotoxic T cells and memory cytotoxic T cells; also, the spatial pattern of these immune cells were characterized for having higher spatial proximity to malignant cells in these patients [53]”.
- Line 141 – 154: “Moreover, integration of multiomic technologies using DNA and RNA approaches techniques, such as whole exome sequencing, whole genome sequencing, targeted sequencing, RNA sequencing, and single cell RNA sequencing, have been used to explore the high complexity of the TME, and can further aid to biomarker discovery. For instance, single cell sequencing in lung adenocarcinoma early-stages has identified KRT8+ alveolar intermediate cells as potential precursors of malignancy, with the integration of whole exome and spatial transcriptomics data [56]. Additionally, single cell RNA sequencing of B cells, and their receptor, revealed an increase in B cells within the TME of lung adenocarcinoma with a notable maturation towards plasma cells, especially among smokers. The abundance of plasma cells was associated with improved survival and better response to immunotherapy, suggesting use the signature of plasma cells as potential biomarker for response of anti-PD-1/PD-L1 therapy [57]. Furthermore, reduction in cytolytic activity related to cell mediators of antitumor immunity was identified using single cell sequencing and validated by flow cytometry and mIF [58]”.
Comment 3: “Sections 3 and 4 need additional examples for support”.
Response: We thank the reviewer for raising this point. To address this, we added examples of the use of the different technologies in tumor biomarker studies, in the following lines:
- Line 203 – 210: “As an example of the utilization of this approach in lung cancer, in one study, highly multiplexed imaging mass cytometry was used to analyze 416 primary lung adenocarcinoma samples from treatment-naïve patients, most of them early stage (I-II, 87.74%). In addition to tumors and endothelial cells, they profiled 14 immune cell types. The study evaluated the correlation between cell frequencies, their interactions with clinical data and the link between survival and cellular phenotypes in the TME, finding that, independently from tumor architecture, the spatial location of the cells predicts survival [70]”.
- Line 218 – 228: “Some studies have been conducted in lung cancer using spatial multi-omic analysis such as the high-plex digital spatial profiling (DSP). For example, in samples from patients enrolled in a phase II multicenter clinical trial (NCT03838848) [patients with advanced NSCLC treated with anti-PD-L1/CTLA-4 bispecific antibody (KN046)], researchers analyzed the impact of intratumoral heterogeneity using spatial proteomic and transcriptomic DSP, and found spatially defined tumor and stromal signatures scores that could predict clinical response [75]. Similarly, using spatial protein profiling, a study of 67 advanced NSCLC samples from patients treated with anti-PD-1 therapy showed ex-pression of VISTA and CD127 in the tumor compartment as markers of immunotherapy resistance, while high CD56 and CD4 were associated with both overall and progression-free survival [76]”.
- Line 299 – 304: “This technique has been used in one study of NSCLC, with integration of single-cell RNA sequencing, revealing diverse phenotypes and gene expression profiles of cancer-associated fibroblasts (CAFs). Notably, a subcluster of myofibroblastic CAFs, POSTN+, was enriched in advanced tumors, exhibiting signatures linked to extracellular matrix remodeling, tumor invasion and immune suppression and hence associated with poor prognosis [105]”.
- Line 343 – 356: “Metabolic reprogramming significantly influences the progression of different types of cancer, these specific metabolic alterations are still not fully understood, limiting their clinical application [113], therefore, to overcome this lack of knowledge, several studies are undergoing on this area. For example, in lung cancer, high levels of glycine, glycolate glyceric acid, creatinine, aminomalonic acid and oxalate were identified as risk factors in a study that used gas chromatography-mass spectrometry to identify plasma metabolites. Oxalate was found to be a unique metabolic risk factor and lactate dehydrogenase A, an enzyme related to oxalate metabolism, was associated with poor prognosis [114]. In this context, RPPA technology, integrated with different analytic platforms, such as mass spectrometry, metabolomics, transcriptional profiling, DNA sequencing and epigenomics, has proven to be particularly useful. [115]. In patients with lung adenocarcinoma, mitochondrial proteins related to fatty acid oxidation, antioxidant response and oxidative phosphorylation, identified with RPPA showed being an independent predictor of survival, encouraging the use of mitochondrial activity as target of therapy [116]”.
Comment 4: “The content in Section 5 is vague and lacks substance”.
Response: We appreciate the reviewer’s comment. To address this issue, we re-wrote this section summarizing findings and adding limitations and future capabilities of the assays. These changes can be found in the following lines:
- Lines 382-455: “Traditionally, the study of lung cancer has relied on the analysis of conventional hematoxylin and eosin slides and low-plex immunohistochemistry [4]. Nowadays, tumors and their microenvironment are the subject of constant discovery, and the understanding of the TME has been revolutionized by bulk single-cell techniques, further enhanced by spatial transcriptomics data and spatial interactions within tumor cells and TME components.
In addition to the proteome, the transcriptome can be useful for designing new thera-pies, identifying genes that are impacted by various genomic and epigenomic changes and the detection of large rearrangements of chromosomes [64]. Integrating diverse levels of analysis including not only spatial proteomics and transcriptomics, but also metabolomics, genomics and epigenomics, can enhance our understanding of the spatial relationship of cells previously identified with single-cell analysis, the correlation with their functional status and the understanding of different biological pathways. This integration allows us to identify biological changes found in the natural history of the development and progression of lung cancer, enabling the collection of more precise data and hence potential tailored treatments [125].
The use of multi-omics techniques can help to address the biological differences between the transcriptome and proteome. Numerous variables, including post-transcriptional regulation, post-translational modifications, protein breakdown, protein-protein interactions, and environmental stressors, can contribute to this discrepancy. Phosphorylation, acetylation, and glycan modification are examples of changes that can affect proteins and impact biological processes including immunological responses, cell signaling and cancer. Understanding and adjusting these changes could therefore result in the discovery of biomarkers or signatures and novel therapeutic approaches [64].
Since the quality of the data generated from spatial methodologies heavily relies on the experimental design, when selecting a platform to profile lung cancer it is essential to consider the scope of the study, to identify which questions the technology is best suited to answer, and to consider each sample’s inherent heterogeneity. Another important point to consider is time, especially when the biological question comprises the response to treatment and resistance mechanisms [126]. Therefore, it is crucial to rigorously assess every aspect of collection and preparation of samples (preanalytical fac-tors), the methodology of the spatial analysis (resolution, capture area, profiling panel), data generation and analysis [127]. Furthermore, to obtain clinically significant findings, the integration of multiomics data requires various strategies, including conceptualization, data integration, prediction models, networks, and pathways.
Beyond the study planning and execution, performance evaluation of the platforms is a matter to be considered. In this context, one comparison study showed moderate to modest concordance of three commercial single-cell spatial transcriptomics techniques with matching bulk RNA sequencing, revealing some issues with cell segmentation and probe sensitivity in lung cancer and mesothelioma samples [128]. Another study that compared imaging spatial transcriptomics assays on tissue microarrays serial sections containing tumor and normal tissue, found that the platforms evaluated were consistent with orthogonal RNA-seq datasets with different cell segmentation, discovery rates and degrees of sub clustering performance for analysis [129]. These individual characteristics of the platforms can become advantages or liabilities for the purpose of our study, and it is crucial to consider them instead of only be guided by the idea of how much data can we produce.
Currently, there are several initiatives to provide best practices for spatial technologies, such as validation and optimization of the assays, and proper normalization strategies, which are also challenging issues since there are no common standards [126,130,131]. Harmonizing the various processes and putting in place quality control measures for the variation in the different tests is essential. These harmonization processes are necessary to guarantee that data produced from several sites can be compared, allowing accurate results [132-134]. Also, it is very important to include a histopathology perspective of these approaches, especially when using image-based methods, where is possible to correlate molecular data with tissue morphology characterization [135]. Moreover, it is essential to also establish workflows with a multidisciplinary team that ensures multi-omics data integration with clinical information, including the nuances in responses to ICIs and targeted therapies. Although some assays offer basic tools for analysis, in general, the complexity of data analysis and interpretation rises with the number of biomarkers and gene information obtained from the samples, requiring advanced bioinformatic skills and the use of specific software for each step of the analysis, being a significant limitation for application in clinical settings [135].
In conclusion, our understanding of lung cancer tumor biology has advanced in re-cent years, thanks to the development of technologies that allow the profiling of tumor and immune cells to identify their unique composition and functional status. Novel multiomic platforms have been developed and incorporated to clinical trials to evaluate tumor heterogeneity, dynamic gene expression, metabolic reprogramming, signaling pathway activation, cell-to-cell interactions, and immune cell programs. A better understanding of lung cancer tumor biology will lead to biomarker breakthroughs and tailored treatments for patients, while future research focuses on longitudinal spatial profiling tracking changes in the immune landscape over the course of disease progression, treatment, and resistance, in addition to spatial effects of metabolic pathways connected to immune cell distribution and niches roles within the lung cancer TME”.
Reviewer 3 Report
Comments and Suggestions for Authors
The manuscript provides a thorough review of spatial multiomics technologies and their application in lung cancer research, highlighting their potential in understanding tumor heterogeneity, immune responses, and biomarker discovery.
While the topic is timely and clinically significant, the manuscript would benefit from more detailed discussions on the clinical translation of these technologies.
Although the manuscript deals with the tumor microenvironment (TME), it does not investigate deeply into how spatial profiling advances our understanding of cellular composition, signaling, or metabolic interactions within the TME. Additionally, challenges in interpreting spatial multiomics data, such as platform variability, lack of standardized pipelines, and potential biases in data analysis, are not adequately addressed.
The conclusion and future directions section requires refinement to better summarize the findings, discuss limitations, and propose future research avenues.
Author Response
Reviewer #3: 
We would like to thank the reviewer for taking the necessary time and effort to review the manuscript. We sincerely appreciate all your valuable comments and suggestions, which helped us in improving the quality of the manuscript.
Comments and Suggestions for Authors:
Comment 1: “While the topic is timely and clinically significant, the manuscript would benefit from more detailed discussions on the clinical translation of these technologies”.
Response: We thank the reviewer for this helpful comment. To address the lack of clinical studies in our manuscript, we added examples in the following sections:
- Line 118 – 130: “In the phase 2 randomized NEOSTAR trial (NCT03158129) conducted to understand the effect of ICIs (neoadjuvant nivolumab or nivolumab plus ipilimumab) in resectable early-stage NSCLC, mIF and other techniques demonstrated that compared to pre-therapy tumor samples, post-tumor samples have increases in tumor T-cell infiltration and immunologic memory [49]. Likewise, correlative analysis of the Lung-MAP S1400I Phase 3 Randomized Clinical Trial (NCT02785952), conducted to study the benefit to adding Ipilimumab to Nivolumab in patients with advanced squamous cell lung carcinoma chemotherapy-pretreated [52], showed that active immune infiltration, measured by multiplex immunofluorescence (mIF) and gene expression profiling in baseline biopsies were linked to a benefit from ICIs. Furthermore, exceptional responders were associated to higher infiltration of cytotoxic T cells and memory cytotoxic T cells; also, the spatial pattern of these immune cells were characterized for having higher spatial proximity to malignant cells in these patients [53]”.
- Line 141 – 154: “Moreover, integration of multiomic technologies using DNA and RNA approaches techniques, such as whole exome sequencing, whole genome sequencing, targeted sequencing, RNA sequencing, and single cell RNA sequencing, have been used to explore the high complexity of the TME, and can further aid to biomarker discovery. For instance, single cell sequencing in lung adenocarcinoma early-stages has identified KRT8+ alveolar intermediate cells as potential precursors of malignancy, with the integration of whole exome and spatial transcriptomics data [56]. Additionally, single cell RNA sequencing of B cells, and their receptor, revealed an increase in B cells within the TME of lung adenocarcinoma with a notable maturation towards plasma cells, especially among smokers. The abundance of plasma cells was associated with improved survival and better response to immunotherapy, suggesting use the signature of plasma cells as potential biomarker for response of anti-PD-1/PD-L1 therapy [57]. Furthermore, reduction in cytolytic activity related to cell mediators of antitumor immunity was identified using single cell sequencing and validated by flow cytometry and mIF [58]”.
Comment 2: “Although the manuscript deals with the tumor microenvironment (TME), it does not investigate deeply into how spatial profiling advances our understanding of cellular composition, signaling, or metabolic interactions within the TME. Additionally, challenges in interpreting spatial multiomics data, such as platform variability, lack of standardized pipelines, and potential biases in data analysis, are not adequately addressed”.
Response: We thank the reviewer for raising this concern. Advances of the understanding of the TME from spatial profiling techniques were added in the following section:
- Line 141-154: “Moreover, integration of multiomic technologies using DNA and RNA approaches techniques, such as whole exome sequencing, whole genome sequencing, targeted sequencing, RNA sequencing, and single cell RNA sequencing, have been used to explore the high complexity of the TME, and can further aid to biomarker discovery. For instance, single cell sequencing in lung adenocarcinoma early-stages has identified KRT8+ alveolar intermediate cells as potential precursors of malignancy, with the integration of whole exome and spatial transcriptomics data [56]. Additionally, single cell RNA sequencing of B cells, and their receptor, revealed an increase in B cells within the TME of lung adenocarcinoma with a notable maturation towards plasma cells, especially among smokers. The abundance of plasma cells was associated with improved survival and better response to immunotherapy, suggesting use the signature of plasma cells as potential biomarker for response of anti-PD-1/PD-L1 therapy [57]. Furthermore, reduction in cytolytic activity related to cell mediators of antitumor immunity was identified using single cell sequencing and validated by flow cytometry and mIF [58]”.
- Line 203-210: “As an example of the utilization of this approach in lung cancer, in one study, highly multiplexed imaging mass cytometry was used to analyze 416 primary lung adenocarcinoma samples from treatment-naïve patients, most of them early stage (I-II, 87.74%). In addition to tumors and endothelial cells, they profiled 14 immune cell types. The study evaluated the correlation between cell frequencies, their interactions with clinical data and the link between survival and cellular phenotypes in the TME, finding that, independently from tumor architecture, the spatial location of the cells predicts survival [70]”.
- Line 218-228: “Some studies have been conducted in lung cancer using spatial multi-omic analysis such as the high-plex digital spatial profiling (DSP). For example, in samples from patients enrolled in a phase II multicenter clinical trial (NCT03838848) [patients with advanced NSCLC treated with anti-PD-L1/CTLA-4 bispecific antibody (KN046)], researchers analyzed the impact of intratumoral heterogeneity using spatial proteomic and transcriptomic DSP, and found spatially defined tumor and stromal signatures scores that could predict clinical response [75]. Similarly, using spatial protein profiling, a study of 67 advanced NSCLC samples from patients treated with anti-PD-1 therapy showed ex-pression of VISTA and CD127 in the tumor compartment as markers of immunotherapy resistance, while high CD56 and CD4 were associated with both overall and progression-free survival [76]”.
- Line 299-304: “This technique has been used in one study of NSCLC, with integration of single-cell RNA sequencing, revealing diverse phenotypes and gene expression profiles of cancer-associated fibroblasts (CAFs). Notably, a subcluster of myofibroblastic CAFs, POSTN+, was enriched in advanced tumors, exhibiting signatures linked to extracellular matrix remodeling, tumor invasion and immune suppression and hence associated with poor prognosis [105]”.
Also, challenges and limitations of the platform performance and data analysis can be found in the conclusions and future directions section:
- Line 398-428: The use of multi-omics techniques can help to address the biological differences between the transcriptome and proteome. Numerous variables, including post-transcriptional regulation, post-translational modifications, protein breakdown, protein-protein interactions, and environmental stressors, can contribute to this discrepancy. Phosphorylation, acetylation, and glycan modification are examples of changes that can affect proteins and impact biological processes including immunological responses, cell signaling and cancer. Understanding and adjusting these changes could therefore result in the discovery of biomarkers or signatures and novel therapeutic approaches [64].
Since the quality of the data generated from spatial methodologies heavily relies on the experimental design, when selecting a platform to profile lung cancer it is essential to consider the scope of the study, to identify which questions the technology is best suited to answer, and to consider each sample’s inherent heterogeneity. Another important point to consider is time, especially when the biological question comprises the response to treatment and resistance mechanisms [126]. Therefore, it is crucial to rigorously assess every aspect of collection and preparation of samples (preanalytical fac-tors), the methodology of the spatial analysis (resolution, capture area, profiling panel), data generation and analysis [127]. Furthermore, to obtain clinically significant findings, the integration of multi-omics data requires various strategies, including conceptualization, data integration, prediction models, networks, and pathways.
Beyond the study planning and execution, performance evaluation of the platforms is a matter to be considered. In this context, one comparison study showed moderate to modest concordance of three commercial single-cell spatial transcriptomics techniques with matching bulk RNA sequencing, revealing some issues with cell segmentation and probe sensitivity in lung cancer and mesothelioma samples [128]. Another study that compared imaging spatial transcriptomics assays on tissue microarrays serial sections containing tumor and normal tissue, found that the platforms evaluated were consistent with orthogonal RNA-seq datasets with different cell segmentation, discovery rates and degrees of sub clustering performance for analysis [129]. These individual characteristics of the platforms can become advantages or liabilities for the purpose of our study, and it is crucial to consider them instead of only be guided by the idea of how much data can we produce.
Comment 3: “The conclusion and future directions section requires refinement to better summarize the findings, discuss limitations, and propose future research avenues”.
Response: To address this issue, we re-wrote this section summarizing findings and adding limitations and future capabilities of the assays. These changes can be found in the following lines:
- Lines 382-455: “Traditionally, the study of lung cancer has relied on the analysis of conventional hematoxylin and eosin slides and low-plex immunohistochemistry [4]. Nowadays, tumors and their microenvironment are the subject of constant discovery, and the understanding of the TME has been revolutionized by bulk single-cell techniques, further enhanced by spatial transcriptomics data and spatial interactions within tumor cells and TME components.
In addition to the proteome, the transcriptome can be useful for designing new thera-pies, identifying genes that are impacted by various genomic and epigenomic changes and the detection of large rearrangements of chromosomes [64]. Integrating diverse levels of analysis including not only spatial proteomics and transcriptomics, but also metabolomics, genomics and epigenomics, can enhance our understanding of the spatial relationship of cells previously identified with single-cell analysis, the correlation with their functional status and the understanding of different biological pathways. This integration allows us to identify biological changes found in the natural history of the development and progression of lung cancer, enabling the collection of more precise data and hence potential tailored treatments [125].
The use of multi-omics techniques can help to address the biological differences between the transcriptome and proteome. Numerous variables, including post-transcriptional regulation, post-translational modifications, protein breakdown, protein-protein interactions, and environmental stressors, can contribute to this discrepancy. Phosphorylation, acetylation, and glycan modification are examples of changes that can affect proteins and impact biological processes including immunological responses, cell signaling and cancer. Understanding and adjusting these changes could therefore result in the discovery of biomarkers or signatures and novel therapeutic approaches [64].
Since the quality of the data generated from spatial methodologies heavily relies on the experimental design, when selecting a platform to profile lung cancer it is essential to consider the scope of the study, to identify which questions the technology is best suited to answer, and to consider each sample’s inherent heterogeneity. Another important point to consider is time, especially when the biological question comprises the response to treatment and resistance mechanisms [126]. Therefore, it is crucial to rigorously assess every aspect of collection and preparation of samples (preanalytical fac-tors), the methodology of the spatial analysis (resolution, capture area, profiling panel), data generation and analysis [127]. Furthermore, to obtain clinically significant findings, the integration of multiomics data requires various strategies, including conceptualization, data integration, prediction models, networks, and pathways.
Beyond the study planning and execution, performance evaluation of the platforms is a matter to be considered. In this context, one comparison study showed moderate to modest concordance of three commercial single-cell spatial transcriptomics techniques with matching bulk RNA sequencing, revealing some issues with cell segmentation and probe sensitivity in lung cancer and mesothelioma samples [128]. Another study that compared imaging spatial transcriptomics assays on tissue microarrays serial sections containing tumor and normal tissue, found that the platforms evaluated were consistent with orthogonal RNA-seq datasets with different cell segmentation, discovery rates and degrees of sub clustering performance for analysis [129]. These individual characteristics of the platforms can become advantages or liabilities for the purpose of our study, and it is crucial to consider them instead of only be guided by the idea of how much data can we produce.
Currently, there are several initiatives to provide best practices for spatial technologies, such as validation and optimization of the assays, and proper normalization strategies, which are also challenging issues since there are no common standards [126,130,131]. Harmonizing the various processes and putting in place quality control measures for the variation in the different tests is essential. These harmonization processes are necessary to guarantee that data produced from several sites can be compared, allowing accurate results [132-134]. Also, it is very important to include a histopathology perspective of these approaches, especially when using image-based methods, where is possible to correlate molecular data with tissue morphology characterization [135]. Moreover, it is essential to also establish workflows with a multidisciplinary team that ensures multi-omics data integration with clinical information, including the nuances in responses to ICIs and targeted therapies. Although some assays offer basic tools for analysis, in general, the complexity of data analysis and interpretation rises with the number of biomarkers and gene information obtained from the samples, requiring advanced bioinformatic skills and the use of specific software for each step of the analysis, being a significant limitation for application in clinical settings [135].
In conclusion, our understanding of lung cancer tumor biology has advanced in re-cent years, thanks to the development of technologies that allow the profiling of tumor and immune cells to identify their unique composition and functional status. Novel multiomic platforms have been developed and incorporated to clinical trials to evaluate tumor heterogeneity, dynamic gene expression, metabolic reprogramming, signaling pathway activation, cell-to-cell interactions, and immune cell programs. A better understanding of lung cancer tumor biology will lead to biomarker breakthroughs and tailored treatments for patients, while future research focuses on longitudinal spatial profiling tracking changes in the immune landscape over the course of disease progression, treatment, and resistance, in addition to spatial effects of metabolic pathways connected to immune cell distribution and niches roles within the lung cancer TME”.
Round 2
Reviewer 2 Report
Comments and Suggestions for Authors
The article has reached the publication standard.